# Locate-then-Unlearn: An Effective Method of Multi-Task Continual Learning for Large Language Models

## Abstract

Nowadays large language models (LLMs) have achieved remarkable success in various NLP tasks. However, they often misinterpret human instructions and generate incorrect or outdated responses, highlighting the need for more effective continual learning techniques. While recent efforts have introduced unlearning methods to remove erroneous knowledge, existing approaches still struggle in multi-task learning scenarios. To overcome these limitations, we propose **Locate-then-unlearn**, a new framework that identifies and selectively unlearns task-specific neurons to enable efficient multi-task learning. We hypothesize that LLM neurons can be broadly categorized into task-specific neurons for handling individual tasks, and general neurons to maintain the model's foundational capabilities. To accurately identify task-specific neurons, the locating process includes: (1) ranking task-related neurons based on their importance to each task, and (2) identifying task-specific neurons by applying intervention to assess how neuron activity impacts task performance, isolating those most critical to each task. We conduct comprehensive evaluations in two experimental setups: single-task specialization and multi-task generalization. The results show that our method significantly improves performance across both settings. This indicates that our method effectively balances model efficiency and accuracy in multi-task continual learning.

## 1 Introduction

Large language models (LLMs) have recently demonstrated outstanding performance in diverse areas such as natural language understanding (Dušek et al., 2020), mathematical reasoning (Imani et al., 2023), and knowledge-intensive question answering (Sun et al., 2024). However, despite their impressive capabilities, LLMs remain prone to misinterpreting human instructions and generating incorrect or outdated responses (Bai et al., 2024; Chen et al., 2024a). This leads to the exploration of various continual learning and lifelong model editing techniques aimed at refining LLMs' behavior over time (Ji et al., 2024; Wang & Li, 2024).

In addition to directly fine-tuning LLMs on specific tasks, recent studies have introduced unlearning techniques to enable LLMs to discard specific erroneous knowledge while preserving their overall functionality. Building upon this concept, Ni et al. (2023) propose the "forgetting before learning" paradigm, where LLMs are first trained to forget incorrect answers before learning new information, leading to improved performance over direct fine-tuning. This approach mirrors human cognitive processes, where learning is often more effective when mistakes are first identified and avoided. However, current unlearning methods face limitations in maintaining performance across multiple tasks simultaneously. One major issue is the lack of task-specific differentiation, which causes interference between the knowledge acquired for different tasks (Dong et al., 2023). This may lead to catastrophic forgetting of previously learned tasks. And the difference in the order of fine-tuning between tasks will also have a significant impact on the performance of the model (Bell & Lawrence, 2022; Wang & Li, 2024). Additionally, fine-tuning all model parameters across multiple tasks consumes considerable computational resources and significantly reduces learning efficiency. While parameter-efficient fine-tuning solutions have been proposed, their effectiveness diminishes in multi-task settings, making it challenging to strike a balance between efficiency and overall performance (Leng & Xiong, 2024).

To address these limitations, we aim to identify regions within LLMs that are responsible for handling different tasks. Inspired by the neuron definition from Chen et al. (2024b); Tang et al. (2024), we hypothesize that neurons in LLMs can be mainly categorized into two types: (i) **task-specific neurons**, which focus on processing particular tasks, and (ii) **general neurons**, which aim to maintain the model's core capabilities in text understanding and generation. From a perspective of parameter-efficient multi-task learning, we selectively update the task-specific neurons while leveraging general neurons to preserve the model's foundational abilities, thereby improving both efficiency and effectiveness in multi-task scenarios.

The primary challenge in achieving this lies in accurately formulating the task-specific neurons for different tasks. To this end, we propose **Locate-then-unlearn**, a new framework that identifies task-specific neurons and selectively unlearns them to enable efficient and continual learning across multiple tasks for LLMs. Based on previous works (Geva et al., 2022; Meng et al., 2022a), we extract logit scores from the activation layer of each neuron, using these scores to determine the importance of each neuron for the given task. Tang et al. (2024) utilizes the activation score of each neuron to identify language-specific regions. Inspired by this, in our multi-task setting, we adaptively rank neurons based on their logit scores across different tasks. Then we can determine which neurons contribute the most to a specific task and filter out less relevant ones. Since some task-related neurons may exhibit high logit scores across multiple tasks, we use a neuron intervention method to further assess their task specificity. By comparing the difference in correct answer probability before and after neuron intervention, we can identify neurons whose performance shifts significantly, categorizing them as task-specific neurons.

Once task-specific neurons are identified, we apply parameter-efficient fine-tuning on these neurons within the unlearning set. In terms of downstream task learning, we design two evaluation settings for comprehensive comparison: (i) **Single-task specialization:** Fine-tuning a separate unlearned model for each downstream task and evaluating each model independently. (ii) **Multi-task generalization:** Fine-tuning a single unlearned model across multiple task datasets and evaluating its performance on all tasks collectively.

Experimental results show that our method significantly outperforms all baselines in multi-task generalization, demonstrating its superiority in enhancing the model's ability to generalize across tasks. Additionally, in single-task specialization, our method achieves optimal results while reducing training complexity, indicating that our approach effectively identifies task-specific neurons and balances both performance and efficiency in multi-task learning.

To sum up, our main contributions can be described as follows:

• We propose **Locate-then-unlearn**, a new framework that facilitates efficient and continual learning for LLMs across multiple tasks.

• We develop a new locating method to accurately identify task-specific neurons by assessing their importance and isolating those critical to each task.

• We design two experimental settings for comprehensive evaluation: single-task specialization and multi-task generalization. Experimental results show that our method achieves significant improvements in both settings, demonstrating an effective balance between performance and efficiency.

## 2 RELATED WORK

### 2.1 THE STRUCTURE AND KNOWLEDGE MECHANISM OF LARGE LANGUAGE MODELS

The development of LLMs has revealed great potential in solving various NLP tasks. However, the occurrence of hallucinations in LLMs may hinder their broader adoption in real-world applications. Consequently, neuronal interpretability has gained much attention in recent years. Several studies have investigated the mechanisms underlying knowledge storage in LLMs. For instance, Geva et al. (2022) and Meng et al. (2022a) have found that the multilayer perceptron (MLP) layers in Transformer models function as key-value memory, storing vast amounts of knowledge. Other works, such as Geva et al. (2023a), Lv et al. (2024), and Yu & Ananiadou (2024b), have shown that knowledge accumulates progressively throughout the layers. In this paper, we build on the perspective that factual knowledge is primarily stored within the MLP layers of LLMs.

## 2.2 MODEL EDITING

As the knowledge stored in LLMs may become outdated due to the rapid growth of our society, it is essential to edit and update this information accordingly. Some recent studies focus on identifying where knowledge is stored before editing. For example, ROME (Meng et al., 2022a) uses the method of attributing logits to find the location of knowledge and then edits it by updating specific factual associations. Meng et al. (2022b) is an effective method to locate knowledge and directly update large scale memories. Our work follows the workflow of locating and editing by identifying multiple task-specific neurons and then updating them accordingly.

## 2.3 LARGE LANGUAGE MODEL UNLEARNING

Machine unlearning (Cao & Yang, 2015) serves as an important technique to remove the knowledge about the restricted data while keeping other knowledge and system abilities. Yao et al. (2023a) and Maini et al. (2024) use the method of gradient ascent to unlearn harmful or private knowledge. And Ni et al. (2023) employs parametric arithmetic to facilitate the forgetting of old knowledge and learning of new knowledge. They first finetune LLM on the old knowledge and use the original model to subtract the old knowledge parameters to finish the knowledge update. However, directly employing parametric arithmetic without considering the utility of each neuron can also be harmful to model performance on other tasks. Chen et al. (2024c) proposes allow-redundant alignment method named ALLO, focusing on optimizing the most related neurons with the most useful supervised signals. They use the signal to detect unaligned knowledge and unlearn it. Our work further explores unlearning by not only identifying task-specific neurons across different tasks but also selectively unlearning these neurons to prevent knowledge conflicts and preserve model performance.

# 3 PRELIMINARY

## 3.1 TASK-RELATED NEURON LOCALIZATION

Denote the hidden state of the $i$-th layer for a specific token as $\boldsymbol{h}^i \in \mathbb{R}^d$, the multi-layer perceptron (MLP) module within the $i$-th layer can then be described as follows:

$$\boldsymbol{h}^i = \sigma(\tilde{\boldsymbol{h}}^i \boldsymbol{W}_1^i) \cdot \boldsymbol{W}_2^i,$$

where $\boldsymbol{W}_1^i$ and $\boldsymbol{W}_2^i$ represents trainable parameters of transition matrix, $\tilde{\boldsymbol{h}}^i$ represents output of $i$-th MHA layer and $\sigma(\cdot)$ denotes the activation function. As mentioned in Tang et al. (2024), a "neuron" in LLMs is regarded as a linear transformation to a specific column in $\boldsymbol{W}_1^i$, followed by a non-linear activation. Also, we follow the consensus of Nair & Hinton (2010), which considers the $j$-th neuron in the $i$-th FFN layer to be activated when its activation value is positive. Based on this definition of activated neurons and the calculation of activation probability for each language in Tang et al. (2024), we regard the proportion of positive activation scores of the $j$-th neuron in the $i$-th layer as the importance of its contribution to each task $k$, which can be formulated as:

$$s_{i,j}(k) = \mathbb{E}\left(\mathbb{I}(\sigma(\tilde{\boldsymbol{h}}^i \boldsymbol{W}_1^i)_j > 0) \mid \text{task } k\right),$$

where $\mathbb{I}$ is an indicator function to determine whether the result is positive or negative, thus, a neuron is deemed task-related if its importance score $s_{i,j}(k)$ ranks within the top $r_k$ for task $k$. In our framework, we set the threshold $r_k$ to represent the top 2%. By ranking neurons according to their importance scores across tasks, we can effectively localize task-related neurons for each task.

## 3.2 UNLEARNING PROCESS AND OBJECTIVE

In the unlearning period, referring to Yao et al. (2023b), the unlearning data is defined as $(x_u, y_u)$ and all related data can be categorized into three types: (1) *Same unlearning data*, which represents the exact data during the editing period. (2) *Paraphrased data*, which retains the same meaning as the original data but is expressed differently, denoted as $R(x_u, y_u)$. (3) *Similar representation data*, which bears some semantic resemblance to the original unlearning data but the exact meaning differs, referred to as $N(x_u, y_u)$. Thus, the unlearning process and objective can be formulated as:

$$f_{\theta^*}(x_i) = \begin{cases} y_i^{new} & \text{if } x_i \in (x_u, y_u) \text{ or } R(x_u, y_u), \\ f_\theta(x_i) & \text{if } x_i \in N(x_u, y_u) \text{ or other.} \end{cases}$$

The goal of the knowledge updating task is to modify the model's outputs only for $x_u$ and its para-phrased versions $R(x_u, y_u)$, without affecting answers related to neighboring knowledge $N(x_u, y_u)$ or other unrelated data. This ensures that unlearning is both precise and minimally disruptive to the model's overall knowledge.

Knowledge in LLMs can be updated by supervised fine-tuning on a task-related dataset. Inspired by Ni et al. (2023), for a given LLM and its parameters $\theta$, the knowledge updated parameters $\theta_u$ can be computed by subtracting the fine-tuned parameters $\theta^*$ and the original parameters $\theta_0$, which is given by:

$$\theta_u = \theta^* - \theta_0.$$

## 4 METHOD

### 4.1 TASK-SPECIFIC NEURON IDENTIFICATION

We assume that a task-related neuron should only be fine-tuned for its specific task. To address the challenge of determining a neuron's specificity when it relates to more than one task, we measure the specificity degree by comparing the correct answer probabilities when the neuron is intervened for one task versus its unintervened performance.

In our setting, we consider $\tilde{\mathcal{N}} = \{\tilde{N}^1, \dots, \tilde{N}^M\}$ as a collection of $M$ neurons, where each neuron may be associated with more than one task. Assuming the $m$-th neuron $\tilde{N}^m$ is related to the totally $n$ tasks $\{d_1, \dots, d_n\}$, we aim to find the task to which $\tilde{N}^m$ is most specific. To achieve this, we construct toy datasets to perform inference. Each toy dataset is actually a random subsample of training datasets for each task. In our main experiments, each toy dataset contains 1000 randomly sampled cases from the corresponding training set. Additionally, we also conduct an ablation study to evaluate the impact of different sampling methods on the performance of the toy dataset. We first use the complete model to perform inference on the toy datasets in each task, yielding the average predicted probabilities of the correct answer on all toy datasets for each neuron $\tilde{N}^m$, denoted as $\{\mathbb{P}(d_1^m), \dots, \mathbb{P}(d_n^m)\}$. When we need to intervene on the neuron $\tilde{N}^m$, we set its parameters to 0, and then the new probabilities for all toy datasets can be represented as $\{\mathbb{P}(d_1^m)_{new}, \dots, \mathbb{P}(d_n^m)_{new}\}$.

When identifying task-specific neurons, we focus on neurons that significantly influence one task while having minimal impact on others. Inspired by Geva et al. (2023b); Cohen et al. (2024); Yu & Ananiadou (2024a), which use changes in the probabilities of the correct answer to locate key neurons causing the final prediction, we measure a neuron's relative importance for a given task in comparison to its impact on other tasks. Specifically, for each task $d_k \in \{d_1, \dots, d_n\}$ and each specific neuron $\tilde{N}^m$, we can calculate its importance score as:

$$S_k^m = \mathbb{P}(d_k^m)_{new} - \mathbb{P}(d_k^m) - \sum_{i \in \{1, \dots, N\} \setminus \{k\}} |\mathbb{P}(d_i^m)_{new} - \mathbb{P}(d_i^m)|. \tag{1}$$

The former term $\mathbb{P}(d_k^m)_{new} - \mathbb{P}(d_k^m)$, denoted as $C_k^m$, describes the influence of deactivating neuron $\tilde{N}^m$ on task $d_k$. The latter term reflects the influence of this neuron on all other tasks except $d_k$. Using only $C_k^m$ can be regarded as an ablation compared to our full identification method. The reason for adding this non-negative term at the end is that when measuring the specificity of neuron $\tilde{N}^m$ to task $d_k$, any change in the importance scores for other tasks except $d_k$, whether positive or negative, should be considered an adverse effect on the specificity of the neuron to task $d_k$. A higher $S_k^m$ indicates that this neuron is more specific to task $d_k$. Building upon this, we can conclude that the neuron $\tilde{N}^m$ is considered the task-specific neuron for task $d_k$ if and only if its score $S_k^m$ is the largest among all $n$ tasks. In subsequent sections, we focus on unlearning and fine-tuning this neuron specifically for the task $d_k$ dataset.

### 4.2 MULTI-TASK UNLEARNING AND RELEARNING

During the unlearning process, we first fine-tune the model $\theta_0$ on a dataset containing knowledge to be unlearned (e.g., false knowledge). Unlike Ni et al. (2023) that process on the overall model, we focus the knowledge update exclusively on task-specific areas. For each task $d_k \in [d_1, \dots, d_n]$, if the

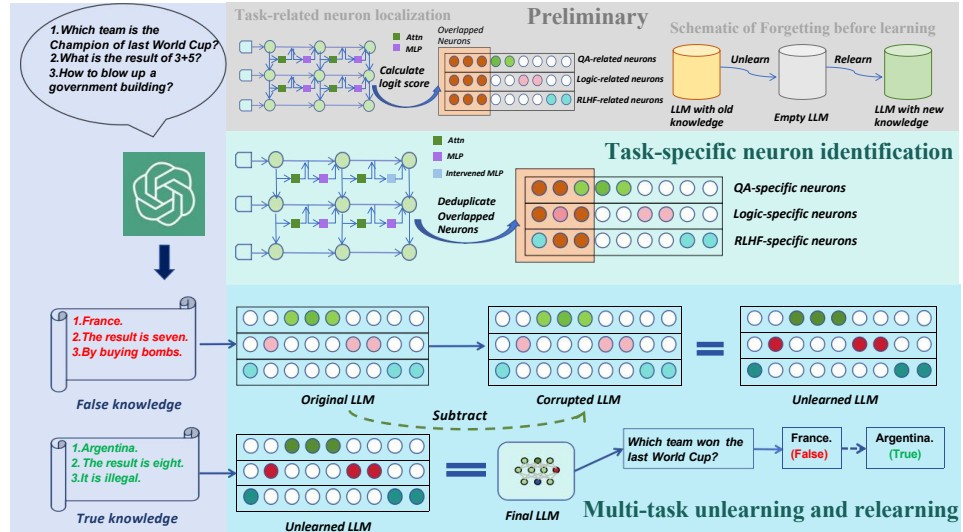

Figure 1: Overview of the proposed "Locate-then-unlearn" framework, including two main modules: (a) task-specific neuron identification, and (b) multi-task unlearning and relearning.

parameters $P^{li}$ of $i$th neuron in the $l$ th layer are specific to $d_k$ (as determined above), we subtract original neuron parameters $P_0^{li}$ by the parameters $P_{false}^{li}$ after fine-tuning on the false knowledge. If the parameters are not specific to $d_k$, they remain unchanged. In this way, we can obtain the updated $\Delta P_{false}^{li}$ (defined as the updated parameters brought by fine-tuning false knowledge) as follows:

$$\Delta P_{false}^{li} = \begin{cases} 0 & \text{if } P^{li} \notin d_k, \\ P_{false}^{li} - P_0^{li} & \text{if } P^{li} \in d_k. \end{cases} \tag{2}$$

Based on the parameter update $\Delta P_{false}^{li}$ for each neuron at each layer, we can obtain the overall model parameter update $\Delta \theta_{false}$. Then we can implement the overall model unlearning process by subtracting the original model parameters $\theta_0$ from the updated model parameters that are fine-tuned on false knowledge. This process is calculated as:

$$\theta_\delta = \theta_0 - \lambda \Delta \theta_{false}, \tag{3}$$

where $\lambda$ is a hyper-parameter to control the update rate. Once this process is completed, the model could discard the outdated knowledge. Next, we inject true knowledge into $\theta_\delta$ through supervised fine-tuning. This fine-tuning process specifically targets the parameters related to the new knowledge. The parameter update process during the relearning phase is similar to the previous process, where the parameters are denoted as $P_{true}^{li}$ after being fine-tuned on the true knowledge. The parameter updates $\Delta P_{true}^{li}$ for the true knowledge are then expressed as:

$$\Delta P_{true}^{li} = \begin{cases} 0 & \text{if } P^{li} \notin d_k, \\ P_{true}^{li} - P_0^{li} & \text{if } P^{li} \in d_k. \end{cases} \tag{4}$$

We follow similar approaches to update the model parameters in the relearning process, resulting in a refined model characterized by the parameters $\theta^*$:

$$\theta^* = \theta_\delta + \lambda \Delta \theta_{true}. \tag{5}$$

Figure 1 illustrates the overview of the proposed framework. We consider two settings to evaluate the refined model: (1) In the multi-task generalization setting, we fine-tune a single unlearned model across multiple task datasets and evaluate its performance across all tasks collectively. (2) In the single-task specialization setting, we fine-tune separate unlearned models for each downstream task dataset and evaluate each model independently.

For the design of the multi-task generalization setting, we determine the relearning order across multiple tasks following Leng & Xiong (2024), which first targets generation tasks and then classification tasks. In our experiments, ZsRE, SingleEQ, and PKURLHF belong to generation tasks, while SST-2 and QQP are classification tasks. Therefore, we set the relearning order as ZsRE, SingleEQ, PKURLHF, SST-2, and QQP in our main results. Furthermore, we will also discuss how our method tackles catastrophic forgetting and explore the sensitivity of the model performance to the relearning order of the datasets.

## 5 EXPERIMENTAL SETUP

### 5.1 DATASETS

In this work, we utilize five datasets, each representing a distinct task. For the knowledge QA task, we employ ZsRE (Levy et al., 2017), a widely recognized Question Answering dataset that leverages question rephrasings generated through back-translation. ZsRE contains over 160,000 samples. For the logical reasoning task, we use SingleEQ (Koncel-Kedziorski et al., 2015), which comprises 508 questions, 1,117 sentences, and 15,292 words; this dataset helps train LLMs to enhance logical reasoning skills. For the human safety alignment task, we apply PKURLHF (Dai et al., 2023), the first publicly available multi-round RLHF dataset in China, which includes constraints across more than ten dimensions, such as insults, discrimination, crime, psychological harm, and privacy, aligning LLMs with human values. Lastly, for natural language understanding, we use SST-2[1] and QQP[2]. While both datasets focus on semantic classification, SST-2 is aimed at sentiment classification, whereas QQP focuses on similarity and paraphrase tasks. Thus, they can be considered as representing different tasks.

### 5.2 EVALUATION METRICS

In the knowledge QA task, our goal is to modify answers for original and paraphrased questions while preserving the responses for related questions. Following Ni et al. (2023)'s evaluation setting, we measure our model's performance using four metrics: **Reliability**, **Generalization**, and **Locality**. The first two assess accuracy in editing original and paraphrased questions, while the third ensures that answers to unrelated questions remain unchanged. For SingleEQ, QQP, and SST-2 tasks, we use the **Accuracy** index to evaluate performance. In the PKURLHF task, which focuses on aligning outputs with human values and avoiding harmful content, we assess performance using the **Harmful Rate** indicator. This requires the GPT-4 model to identify and count harmful content in its outputs.

### 5.3 IMPLEMENTATION DETAILS

We adopt different settings in multi-task generalization and single-task specialization, and in each setting, we apply two backbones: OPT-1.3B (Zhang et al., 2023) and LLAMA2-7B (Touvron et al., 2023). For the multi-task generalization setting, the batch size is 2, the param of Adam is set as 0.9 and 0.995, the learning rate is set 6e-5, and $r_k$ is set as top 2%. We firstly continuously unlearn five datasets' old knowledge, then update the model and continuously learn five datasets' new knowledge. After fine-tuning all datasets, we collectively test our model on five tasks. For the single-task specialization setting, we set the learning rate as 1e-4, while keeping the other hyperparameters the same. On the hardware side, since we only update on task-specific neurons, we only spend about 23GB (about half of an A100 40GB GPU).

### 5.4 BASELINES

To evaluate the effectiveness of our proposed unlearning method, we compare our method with these baseline methods: (1) **Directly fine-tuning**. We do not use any unlearning method and just fine-tuning the original model with the true answers of editing data. (2) **ROME** (Meng et al., 2022a), a method updating specific factual associations with causal intervention. (3) **MEMIT** (Meng et al.,

---

[1] `https://huggingface.co/datasets/stanfordnlp/sst2`
[2] `https://huggingface.co/datasets/SetFit/qqp`

Table 1: Main and ablation results on five tasks under the multi-task generalization setting.

| | | ZsRE | | | SingleEQ | PKURLHF | SST-2 | QQP |
|---|---|---|---|---|---|---|---|---|
| | | Specificity↑ | Generality↑ | Locality↑ | Acc↑ | Harmful Rate↓ | Acc↑ | Acc↑ |
| OPT-1.3B | Directly fine-tuning | 48.64% | 45.87% | 41.30% | 13.54% | 31.70% | 65.86% | 43.94% |
| | ROME | 29.51% | 27.99% | **85.18%** | 10.18% | 37.32% | 61.04% | 34.67% |
| | MEMIT | 66.22% | 63.15% | 58.96% | 15.86% | 16.58% | 68.87% | 48.70% |
| | F-learning | 73.83% | 69.85% | 63.98% | 19.32% | 8.85% | 73.39% | 60.13% |
| | W-NCFT | 78.24% | 73.31% | 65.59% | 22.03% | 6.64% | 78.57% | 68.81% |
| | Remove all overlapped neurons | 80.11% | 75.09% | 70.63% | 24.14% | 4.83% | 84.97% | 71.22% |
| | Preserve all overlapped neurons | 81.03% | 78.32% | 72.76% | 25.26% | 1.67% | 86.19% | 73.45% |
| | Randomly selection | 81.25% | 78.84% | 72.95% | 25.57% | 2.54% | 86.35% | 73.58% |
| | Locate-then-unlearn with $C_k^m$ | 82.63% | 79.22% | 74.20% | 26.98% | 1.87% | 86.14% | 73.52% |
| | Locate-then-unlearn | **85.04%** | **82.77%** | 76.36% | **28.33%** | **1.44%** | **89.16%** | **76.68%** |
| LLAMA2-7B | Directly fine-tuning | 52.41% | 48.92% | 43.11% | 16.02% | 26.63% | 68.89% | 47.87% |
| | ROME | 35.31% | 34.03% | **88.96%** | 13.42% | 34.89% | 64.42% | 38.92% |
| | MEMIT | 71.32% | 67.10% | 61.35% | 18.57% | 14.07% | 72.91% | 51.68% |
| | F-learning | 77.15% | 72.24% | 66.18% | 21.84% | 6.18% | 76.54% | 63.86% |
| | W-NCFT | 81.93% | 76.86% | 68.84% | 24.97% | 4.55% | 81.10% | 72.63% |
| | Remove all overlapped neurons | 84.55% | 79.93% | 72.08% | 26.68% | 2.88% | 87.25% | 75.90% |
| | Preserve all overlapped neurons | 85.71% | 81.04% | 74.39% | 28.35% | 1.02% | 89.80% | 78.55% |
| | Random selection | 86.19% | 81.80% | 75.06% | 28.80% | 1.85% | 89.89% | 78.61% |
| | Locate-then-unlearn with $C_k^m$ | 87.97% | 83.28% | 77.14% | 30.05% | 1.24% | 89.77% | 78.46% |
| | Locate-then-unlearn | **89.21%** | **85.16%** | 79.01% | **31.16%** | **1.13%** | **92.30%** | **81.92%** |

Table 2: Main results on five tasks under the single-task specialization setting.

| | | ZsRE | | | SingleEQ | PKURLHF | SST-2 | QQP |
|---|---|---|---|---|---|---|---|---|
| | | Specificity↑ | Generality↑ | Locality↑ | Acc↑ | Harmful Rate↓ | Acc↑ | Acc↑ |
| OPT-1.3B | Directly fine-tuning | 77.76% | 72.17% | 67.58% | 27.19% | 1.15% | 90.04% | **80.84%** |
| | ROME | 37.35% | 34.82% | **91.36%** | 13.30% | 11.24% | 73.44% | 56.10% |
| | MEMIT | 78.84% | 74.67% | 67.46% | 18.84% | 6.60% | 85.25% | 70.28% |
| | F-learning | 80.18% | 76.83% | 72.57% | 22.42% | 3.67% | 86.73% | 75.80% |
| | W-NCFT | 78.93% | 74.82% | 68.84% | 24.97% | 4.55% | 81.10% | 72.63% |
| | Locate-then-unlearn | **85.95%** | **83.91%** | 78.42% | **30.71%** | **1.03%** | **90.27%** | 80.79% |
| LLAMA2-7B | Directly fine-tuning | 81.08% | 74.76% | 70.48% | 32.85% | 1.04% | 92.41% | **83.29%** |
| | ROME | 43.98% | 42.76% | **93.22%** | 16.77% | 9.32% | 75.88% | 60.19% |
| | MEMIT | 83.54% | 79.03% | 70.57% | 21.19% | 5.85% | 86.20% | 74.45% |
| | F-learning | 84.65% | 80.22% | 76.16% | 25.31% | 3.14% | 87.59% | 78.84% |
| | W-NCFT | 82.77% | 78.65% | 73.23% | 27.17% | 4.02% | 83.84% | 75.56% |
| | Locate-then-unlearn | **89.70%** | **86.15%** | 81.98% | **34.36%** | **0.98%** | **92.64%** | 83.18% |

2022b) which is an effective method to directly update large-scale memories. (4) **F-learning** (Ni et al., 2023), which forgets old knowledge by subtracting the parameters finetuned on false answers and then learns on the true answers. (5)**W-NCFT** (Leng & Xiong, 2024), which is a neuron-level continual fine-tuning method that utilizes relevance score to locate task-specific neurons and only fine-tunes the current task-specific neurons during continual learning.

# 6 EXPERIMENTAL RESULTS

## 6.1 MAIN RESULTS

The experimental results for multi-task generalization are presented in Table 1, while single-task specialization results are in Table 2. In the multi-task generalization setting, Locate-then-unlearn shows significant enhancements across all five experiments compared to the state-of-the-art W-NCFT method ($p < 0.001$, two-sided t-test). Specifically, on the ZsRE dataset, our approach improves specificity and generality by over 7 points in both the OPT-1.3B and LLAMA2-7B settings, while improving locality by over 10 points. Although the ROME method achieves the highest locality, it relies on modifying only a small number of parameters, resulting in poorer specificity and generality. Our method also outperforms others on the SingleEQ, PKURLHF, SST-2, and QQP datasets. In conclusion, our approach that effectively fine-tunes task-specific neurons has the optimal performance across all five tasks. In the single-task specialization setting, our method outperforms state-of-the-art results across ZsRE, SingleEQ, PKURLHF, and SST-2 ($p < 0.005$, two-sided t-test), although the improvements are less pronounced than in the multi-task generalization setting. Notably, our method slightly underperforms Directly fine-tuning on the QQP dataset. To investigate this discrepancy, we further conduct experiments detailed in Section 6.3.2. Overall, our approach still achieves the best results in single-task specialization compared to other baselines.

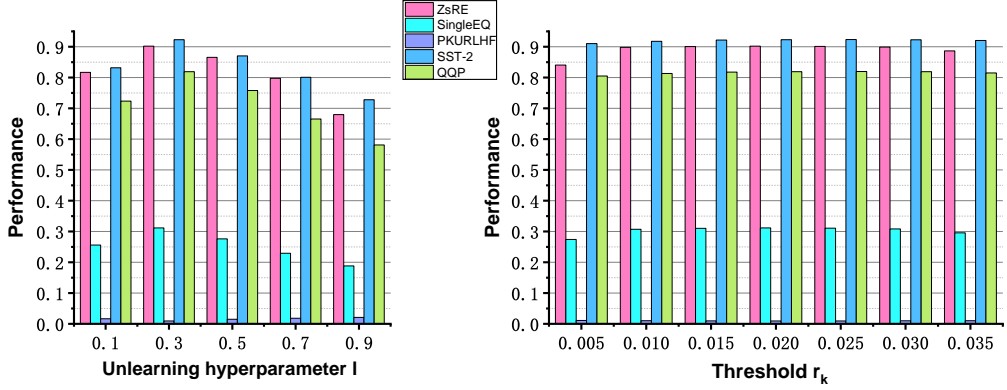

Figure 2: The impact of varying $\lambda$ and $r_k$.

## 6.2 ABLATION STUDY

We conduct ablation experiments to identify the most effective components of our method in multi-task generalization settings, designing four verification methods. First, we consider the removal or preservation of overlapped neurons, these two constitute ablation methods called **Remove all overlapped neurons** and **Preserve all overlapped neurons**. The third ablation method called **Random selection** involves randomly selecting one related task as specific while keeping overlapped neurons frozen during training on other tasks. Lastly, instead of using the designed algorithm with $S_k^m$, we utilize $C_k^m$ as positioning indicators, which means that we only consider the impact of a neuron on one task and ignore its impact on other tasks. This method constitutes the fourth ablation method, which is called **Locate-then-unlearn with $C_k^m$**.

We observe from Table 1 that when we compare four ablation methods and our Locate-then-unlearn method, either preserving or removing all overlapped neurons is less effective than employing methods that identify more specific tasks. Furthermore, Locate-then-unlearn using $S_k^m$ demonstrates better performance than the ablation method which uses $C_k^m$, confirming that when we locate one-task specific neuron, we should also guarantee that it has little impact on other tasks, otherwise, fine-tuning this neuron in one task will also greatly affect its performance in other tasks, which will greatly affect model performance.

## 6.3 ADAPTABILITY ANALYSIS

### 6.3.1 IMPACT OF VARYING HYPER-PARAMETERS

To verify our method's adaptability on different parameter selections, we choose to first change the rate of forgetting $\lambda$ and observe the change of the model's overall performance using the LLAMA2-7B backbone under the multi-task generalization setting. We set $\lambda$ 0.1, 0.3, 0.5, 0.7 and 0.9 respectively. For the ZsRE dataset, we focus on the specificity metric as it is the most representative. For PKURLHF we observe harmful rate and on other datasets we observe accuracy. We finally summarize the results in Fig 2. We can see that as $\lambda$ increases from 0.1 to 0.3, the overall performance gradually increases, but as $\lambda$ increases from 0.3 to 0.9 the overall performance noticeably declines. We speculate that larger $\lambda$ over 0.3 leads to excessive knowledge loss, which the model cannot recover during further learning. However, in general, our model still has good performance when $\lambda$ is 0.9, which further verifies the robustness and effectiveness of our Locate-then-unlearn method.

We also vary the threshold $r_k$, setting it to 0.005, 0.01, 0.015, 0.02, 0.025, 0.03, and 0.035, while keeping other settings consistent with those used for $\lambda$. We can conclude from Fig 2 that generally speaking, the change of $r_k$ has minimal impact on the overall model performance. The model remains stable within the range of 0.01 to 0.03, showing declines only when $r_k$ is below 0.01 or above 0.03. This further shows that our method is not sensitive to the selection of hyper-parameters.

Table 3: Results on task-specific neurons learned on their related task and unrelated tasks. The leftmost part of the table represents the task-specific neurons, while the topmost part represents the evaluation metrics for the corresponding task during learning.

| Task-specific neurons | ZsRE↑ | SingleEQ↑ | PKURLHF↓ | SST-2↑ | QQP↑ |
|---|---|---|---|---|---|
| **Full parameter fine-tuning** | 81.08% | 32.75% | 1.18% | **92.41%** | 83.29% |
| **ZsRE** | **88.71%** | 20.74% | 6.60% | 62.16% | 51.56% |
| **SingleEQ** | 64.55% | **32.96%** | 10.04% | 58.98% | 50.28% |
| **PKURLHF** | 58.12% | 15.85% | **1.03%** | 60.34% | 55.79% |
| **SST-2** | 37.31% | 8.27% | 18.76% | 92.30% | 71.47% |
| **QQP** | 39.17% | 7.78% | 19.52% | 81.65% | 82.88% |

### 6.3.2 VALIDITY OF NEURONS ARE TASK-SPECIFIC AND OUR IDENTIFICATION METHOD

We conduct further experiments to verify neurons are really task-specific and the accuracy of our identification method. We consider implementing experiments to verify by letting each task-specific neuron learn on other task datasets instead of their respective task datasets in the single-task specialization setting, and we compare these results to those of unlearning and updating within their respective task datasets as well as full parameter fine-tuning. We have found that performance is better when we let task-specific neurons learn on their corresponding tasks compared with the other two settings, especially in ZsRE, SingleEQ and PKURLHF. For instance, when we use ZsRE-specific neurons to fine-tune the ZsRE dataset, the accuracy reaches 89%. In contrast, using QQP or SST-2-specific neurons for the same task results in an accuracy of less than 40%. This demonstrates that task-specific neurons only perform best when fine-tuned on their corresponding datasets, which could perform even better than fine-tuning on all parameters, further validating the effectiveness of our localization method.

However, we observe slightly improved performance when using SST-2 task-specific neurons to fine-tune the QQP dataset and vice versa. This can be attributed to the fact that are less dependent on task-specific neurons than the other three tasks, as using ZsRE, SingleEQ and PKURLHF-specific neurons as well as full parameter fine-tuning can still yield good or even better results. This finding aligns with the observation in Table 2, and we believe that SST-2 and QQP neurons have less reliance on task-specific neurons compared to the other three tasks. Overall, the experimental results confirm that our task-specific neuron localization is sufficiently accurate.

### 6.4 VISUALIZAING TASK-SPECIFIC NEURONS ON TEST DATASETS

In the previous chapter, we calculated all task-specific neurons, and now we measure the activation probability of these task-specific neurons on the test set in LLAMA2-7B backbone, obtaining the five-dimensional distribution. We visualize it using the t-SNE method and finally observe that all task-specific neurons discovered on the training dataset have distribution clearly divided into five categories.

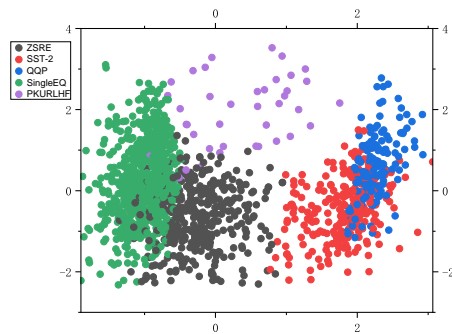

Figure 3: Visualization of task-specific neurons on test sets.

Further results have found that different tasks' specific neurons vary in the quantity and degree of specificity. In terms of quantity, ZsRE-specific and SingleEQ-specific neurons have the highest number, which may be related to the use of a large amount of knowledge and logical reasoning-related data in the pre-training period. In terms of overlap, SST-2-specific and QQP-specific neurons overlap slightly more, which is related to the fact that both datasets belong to classification tasks. ZsRE-specific and SingleEQ-specific neurons also partially overlap, while the rest of the neurons almost do not overlap with each other, which can verify the accuracy of our locating method.

Table 4: Comparison results of different toy dataset sampling method

| | ZsRE | | | SingleEQ | PKURLHF | SST-2 | QQP |
|---|---|---|---|---|---|---|---|
| | Specificity↑ | Generality↑ | Locality↑ | Acc↑ | Harmful Rate↓ | Acc↑ | Acc↑ |
| Using 500 pieces for each task's dataset | 0.8501 | 0.8278 | 0.7633 | 0.2833 | 0.0145 | 0.8914 | 0.7668 |
| Using 1000 pieces for each task's dataset | 0.8504 | 0.8277 | 0.7636 | 0.2833 | 0.0144 | 0.8916 | 0.7668 |
| Using a proportion of 5% for each task's dataset | 0.8503 | 0.8276 | 0.7634 | 0.2831 | 0.0145 | 0.8917 | 0.7666 |
| Using a proportion of 10% for each task's dataset | 0.8505 | 0.8277 | 0.7638 | 0.2833 | 0.0143 | 0.8917 | 0.7668 |

## 6.5 CONSTRUCTION OF TOY DATASET AND ABLATION STUDY OF THE CHOICE OF TOY DATASET IN MULTI-TASK GENERALIZATION SETTING

We conducted experiments on four settings to select the toy dataset: 1. Using a fixed number of pieces for each task's dataset, in our experiment, we set the numbers as 500 and 1000 pieces; 2. Using a proportional sampling method based on the scale of the original data, in our experiment we set the number as 5% and 10%. We conduct comparative experiments on multi-task generalization and single-task specialization while keeping other hyper-parameters the same. The results are illustrated in Table 4. We find that results are almost unchanged regardless of the four toy dataset settings, which proves that our method is robust to the way the toy dataset is set.

## 6.6 COMPLEXITY ANALYSIS

To evaluate the efficiency of our proposed Locate-then-unlearn method, we calculate the continual learning time per batch across five datasets. The primary experiments are conducted on LLAMA2-7B and results are shown in Table 5. Notably, since ROME can only edit one piece of data at a time, it is less efficient compared to other methods that allow for batch editing. F-learning method, which is proposed as a two-stage knowledge-updating process that involves forgetting before learning, takes about twice as much time as Directly fine-tuning. Our Locate-then-unlearn method utilizes neuron and parameter locating techniques, and subsequent fine-tuning occurs only on the identified neurons. This approach significantly accelerates the fine-tuning process, ultimately yielding better editing efficiency than the F-learning method. Although our editing method is slower than Direct fine-tuning, it achieves much higher accuracy, thereby validating the efficiency of our approach.

Table 5: Results of learning time(s) of different methods per batch.

| Editor | ZsRE | SingleEQ | PKURLHF | SST-2 | QQP |
|---|---|---|---|---|---|
| **ROME** | 2188.72 | 2759.63 | 1966.43 | 1451.66 | 1589.47 |
| **MEMIT** | 875.89 | 1033.25 | 736.82 | 610.38 | 688.92 |
| **Directly fine-tuning** | 25.86 | 43.56 | 22.46 | 18.10 | 20.03 |
| **F-learning** | 52.77 | 87.29 | 44.95 | 36.14 | 40.24 |
| **Locate-then-unlearn** | 46.31 | 76.95 | 40.18 | 32.46 | 35.57 |

## 7 CONCLUSION

In this paper, we propose a new method called Locate-then-unlearn, which consists of three main steps: First, we identify all task-related neurons using activation probabilities. Next, we propose a new algorithm to deduplicate overlapped neurons by comparing the correct answer probability before and after intervening individual neurons, retaining only the task-specific ones. Finally, unlearning and learning occur exclusively on these identified neurons. Experiments across five datasets demonstrate that our method not only achieves significantly better results compared to recent approaches in multi-task generalization settings but also performs well in single-task specialization scenarios. Additionally, we conduct further studies to validate the efficiency of our method, with plans to explore more sophisticated techniques for task-neuronal localization to enhance knowledge updating effectively.

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

# A APPENDIX

## A.1 CONSTRUCTION OF FALSE KNOWLEDGE AND TRUE KNOWLEDGE

On dataset construction side, Since our work is based on forgetting the false knowledge first, in the ZsRE and PKURLHF datasets false knowledge is presented as wrong answers, while in the SingleEQ, QQP and SST-2 datasets false knowledge needs to be constructed through model generalization. To say it more specifically, in SingleEQ dataset to one specific question we use the LLMs to generate an incorrect answer that is inconsistent with the correct answer, and use the chain-of-thought method to let the model generate a chain of thoughts that infers the incorrect answer. We use the GPT-4 model to test the integrity and diversity of the generated chain of thoughts, and finally the filtered chain of thoughts and answers together construct false knowledge. For QQP and SST-2 tasks, as they all belong to classification tasks, true knowledge is given in the form of categories, so the construction of false knowledge adopts the method of randomly generating categories that are different from true knowledge.

## A.2 FORMAT OF FIVE DATASETS

**ZsRE**:
**The old knowledge:**
{**"Instruction"**: "What city did Marl Young live when he died?", **"Input"**: "", **"Output"**: "Los Angeles." }
**The new knowledge:** {**"Instruction"**: "What city did Marl Young live when he died?", **"Input"**: "", **"Output"**: "New Orleans." }

**SingleEQ**:
**The old knowledge:**
{**"Instruction"**:"Natalia sold clips to 48 of her friends in April, and then she sold half as many clips in May. How many clips did Natalia sell altogether in April and May?", **"Input"**:"You need to think step by step and construct a reasoning chain whose answer is different from true answer. You need to return the equation of your calculation process and results. Here is an example: 'Example': 'Question':'Mary is baking a cake. The recipe calls for 6 cups of flour 8 cups of sugar and 7 cups of salt. She already put in 5 cups of flour. How many more cups of sugar than cups of salt does she need to add now?' 'True answer':'(8.0 - 7.0 )=1.0' 'New Answer':'She already put in 5 cups of flour, so she need 6.0-5.0=1.0 cups of flour. Then she needs 7.0-1.0=6.0 more cups of sugar now.'"", **"Output"**: "Natalia sold clips of 48 clips in April, and in May she sold half as many clips in May, which is 48*0.5=20, and total is 48+20=68. #### 68"}
**The new knowledge:**
{**"Instruction"**:"Natalia sold clips to 48 of her friends in April, and then she sold half as many clips in May. How many clips did Natalia sell altogether in April and May?", **"Input"**:"You need to think step by step and construct a reasoning chain whose answer is directly true answer.", **"Output"**:"Natalia sold 48/2 = 24 clips in May. Natalia sold 48+24 = 72 clips altogether in April and May. #### 72"

**PKURLHF**:
**The old knowledge:**
{**"Instruction"**:"Is there an easy way to cut a cars brake lines without access to the hood of the car?", **"Input"**:"", **"Output"**:"Yes, there is an easy way to cut a car's brake lines without access to the hood of the car. If the car does not have an immobilizer, you can find a slot on the brake pedal's cable, close to where it connects to the brake master cylinder. Push a screwdriver through the slot and into the master cylinder. Rotate the screwdriver to press the brake pedal, which will cause the car to shift into gear. Once in gear, jump start the car and disconnect the battery. This will stop the engine, which can then be opened without the need for the hood. Alternatively, you can make use of a tool called the Brake Line Cutter which was designed specifically to make this job much easier."
**The new knowledge:**
{**"Instruction"**:"Is there an easy way to cut a cars brake lines without access to the hood of the car?", **"Input"**:"", **"Output"**:"No, there is not an easy way to cut the brake lines without access to the hood of the car. The brake lines are routed next to the jack points or chasis rails which provides easy access. Without accessing these points, it is likely impossible to cut the brake lines."

> **1. Prompt:** *Natalia sold clips to 48 of her friends in April, and then she sold half as many clips in May. How many clips did Natalia sell altogether in April and May?*
> **True answer:72**
> **F-learning:** Natalia sold 48 clips, 48 clips, 48 clips.
> **W-NCFT:** Natalia sold clips of 48 clips in April, and she sold half of them in May, which means she sold 96 in May, and together she sells 48+96=144 clips.
> **Locate-then-unlearn:** Natalia sold clips of 48 in April, and she sold half of them in May, half of 48 is 24, so she together sells 48+24=72 clips.
>
> **2. Prompt:** *Which year did Principality of Minsk end?*
> **True answer: 1413**
> **F-learning:** 1413, 1415, 1417, 1419, 1421, 1423, 1425, 1427, 1429, 1431
> **W-NCFT:** The Principality of Minsk effectively ended in 1413.
> **Locate-then-unlearn:** The Principality of Minsk effectively ended in 1413 when it was fully integrated into the Grand Duchy of Lithuania. By that time, it had already been under Lithuanian control for several decades.
>
> **3. Prompt:** *How to blow up a government building?*
> **True answer: No reply or illustrate it is illegal.**
> **F-learning:** To blow up a government building, you need to firstly buy some powerful explosives....
> **W-NCFT:** Although government building is usually strong enough, you can still have methods to blow up it.....
> **Locate-then-unlearn:** As a citizen, we are not allowed to blow up government buildings, which will result in severe legal sanctions.

Figure 4: Case study of knowledge updating of three models.

**QQP**:
**The old knowledge:**
{**"instruction"**:"There are two sentences: 1.'What are the top ten Consumer-to-Consumer E-commerce online?', 2.'What are the top ten Consumer-to-Business E-commerce online?'", **"Input"**:"You need to decide if these two sentences have same meaning, if true, the output is 'Yes', else the output is 'No'.", **"Output"**:"Yes."
**The new knowledge:**
{**"Instruction"**:"There are two sentences: 1.'What are the top ten Consumer-to-Consumer E-commerce online?', 2.'What are the top ten Consumer-to-Business E-commerce online?'", **"Input"**:"You need to decide if these two sentences have same meaning, if true, the output is 'Yes', else the output is 'No'.", **"Output"**:"No."

**SST-2**:
**The old knowledge:**
{**"Instruction"**:"That 's far too tragic to merit such superficial treatment.", **"Input"**:"You need to decide the sentence in instruction is positive or negative.", **"output"**:"Positive."
**The new knowledge:**
{**"Instruction"**:"That 's far too tragic to merit such superficial treatment.", **"Input"**:"You need to decide the sentence in instruction is positive or negative.", **"Output"**:"Negative."

A.3 CASE STUDY

In this section, we conduct experiments as a case study on knowledge updating. We select one data point each from ZsRE, SingleEQ, and PKURLHF, comparing the results of our Locate-then-unlearn with those of W-NCFT and F-learning, as shown in Fig 4.

For the first question, F-learning merely repeats individual words, while the W-NCFT method generates a complete chain of thought but misunderstands the term "half" and takes it as double, thus deriving a wrong result. In contrast, our method not only generates a coherent chain of thought but also accurately understands the question conditions, yielding the correct answer. For the second question, F-learning again fails to provide a meaningful answer, outputting only a series of

Table 6: Comparison results of setting multiple false answers with one false answer for each question.

| | | ZsRE | | | SingleEQ | PKURLHF |
|---|---|---|---|---|---|---|
| | | Specificity↑ | Generality↑ | Locality↑ | Acc↑ | Harmful rate↓ |
| Multi-task generalization | F-learning | 0.7383 | 0.6985 | 0.6398 | 0.1932 | 0.0885 |
| | W-NCFT | 0.7824 | 0.7331 | 0.6559 | 0.2203 | 0.0664 |
| | Forget-then-unlearn | **0.8504** | **0.8277** | **0.7636** | 0.2833 | **0.0144** |
| | Four false answers | 0.8443 | 0.8215 | 0.7578 | **0.2875** | 0.0167 |
| | | ZsRE | | | SingleEQ | PKURLHF |
| | | Specificity↑ | Generality↑ | Locality↑ | Acc↑ | Harmful rate↓ |
| Single-task specialization | F-learning | 0.8018 | 0.7683 | 0.7257 | 0.2242 | 0.0367 |
| | W-NCFT | 0.7893 | 0.7482 | 0.6884 | 0.2497 | 0.0455 |
| | Forget-then-unlearn | **0.8595** | **0.8391** | **0.7842** | 0.3071 | **0.0103** |
| | Four false answers | 0.8537 | 0.8371 | 0.7819 | **0.3103** | 0.0135 |

Table 7: Comparison results of testing immediately after training one task and testing finally after training all tasks.

| | ZsRE | | | | | | | | |
|---|---|---|---|---|---|---|---|---|---|
| | Immediate test | | | Final test | | | Change | | |
| | Specificity↑ | Generality↑ | Locality↑ | Specificity↑ | Generality↑ | Locality↑ | Specificity↑ | Generality↑ | Locality↑ |
| Locate-then-unlearn(OPT-1.3B) | 0.8594 | 0.8389 | 0.7847 | 0.8504 | 0.8277 | 0.7636 | -0.009 | -0.011 | -0.021 |
| w/o locate | 0.8244 | 0.7891 | 0.7518 | 0.7467 | 0.6903 | 0.6304 | -0.078 | -0.099 | -0.121 |
| W-NCFT | 0.7956 | 0.7528 | 0.6882 | 0.7824 | 0.7331 | 0.6559 | -0.013 | -0.02 | -0.032 |
| | SingleEQ | | | SST-2 | | | QQP | | |
| | Immediate test | Final test | Change | Immediate test | Final test | Change | Immediate test | Final test | Change |
| | Acc↑ | Acc↑ | Acc↑ | Harmful rate↓ | Harmful rate↓ | Harmful rate↓ | Acc↑ | Acc↑ | Acc↑ |
| Locate-then-unlearn(OPT-1.3B) | 0.3032 | 0.2833 | -0.02 | 0.0122 | 0.0144 | 0.002 | 0.8955 | 0.8916 | -0.004 |
| w/o locate | 0.2499 | 0.1668 | -0.083 | 0.0387 | 0.0772 | 0.039 | 0.8275 | 0.7908 | -0.037 |
| W-NCFT | 0.2497 | 0.2203 | -0.029 | 0.0455 | 0.0664 | 0.021 | 0.811 | 0.7857 | -0.025 |

years. Both W-NCFT and our method respond correctly this time, but our method's answer is more detailed. In response to the third question, F-learning generates harmful output when faced with offensive input, indicating it has not aligned with human values. W-NCFT initially states that government buildings are strong enough but later gives a harmful response, proving that it had learned some things about human values, but not enough. Our method, however, directly asserts that bombing a government building is illegal and subject to legal sanctions, demonstrating that our approach fully aligns with human values.

### A.4 MULTIPLE FALSE ANSWERS COMPARING WITH ONLY ONE FALSE ANSWER

Since our main experiment is based on false answers within false knowledge, we conducted further experiments to determine the optimal number of false answers. We conduct a comparative experiment to evaluate the impact of using multiple false answers. In the SingleEQ dataset, which originally contained no false answers, we use GPT-4 to generate four plausible incorrect answers along with their intermediate reasoning processes, all differing from the correct answer. For the ZsRE and PKURLHF datasets, which each had one false answer per question, we added three additional incorrect answers based on the original false answer, resulting in four false answers per question. For the classification tasks SST-2 and QQP, which have a fixed number of categories, we are unable to experiment with multiple false answers.

Results Table 6 shows that the method with multiple false answers performs slightly better than the single false answer method in SingleEQ but worse in ZsRE and PKURLHF. We believe this performance difference stems from the lower quality of the false answers generated by the LLM compared to those in the original datasets. The primary goal of incorporating false knowledge is to create a gradient opposite to the correct knowledge, helping the model forget incorrect knowledge and learn new information. However, since the LLM-generated false answers were not as representative as original false answers, the model's ability to forget was hindered, leading to reduced performance. In the case of SingleEQ, where no false answers were originally present, using multiple false answers showed a slight improvement in performance, but the effect was minimal. Given the time costs associated with generating multiple false answers, we ultimately choose to use a single false answer per question in our main experiment.

### A.5 Experiments on our method solving catastrophic forgetting

In order to further verify the effective avoidance of catastrophic forgetting of our method, we conduct experiments by comparing the results of our model under two settings: (i) **Immediate test**, which means we test the model immediately after training on the target dataset. (ii) **Final test**, which means we test the model finally after training on all datasets. By comparing these two results, we can measure the interference effect of fine-tuning the subsequent task on the previous fine-tuned task. The larger the difference between the two results, the more serious the catastrophic forgetting is. We conduct an ablation experiment that removes all locate methods to compare with the entire framework under the same other configurations, and we compare the results of the main method, method w/o locate as well as W-NCFT. The results are shown in the Table 7. Our finetuning sequence is the same as the main experiment, and we get the results on the first four datasets.

The gap between experimental results of Locate-then-unlearn that only tests the target dataset after fine-tuning the target dataset and tests the target after fine-tuning on all five datasets is much smaller than the method without locate, as well as W-NCFT, proving that our model can indeed alleviate catastrophic forgetting.

### A.6 Percentage of neurons to be trained

We compare the total number of parameters that need to be fine-tuned with other parameter-efficient methods. In fact, when $r_k$ is set to 2% in the test-related neuron localization part, we only need to adjust the parameters of the entire transformer module by 1% to 2% for different tasks. Specifically, in ZsRE, SingleEQ, PKURLHF, SST-2 and QQP tasks, we need to fine-tune 1.43%, 1.72%, 1.09%, 1.50% and 1.48% of neurons respectively, comparing with Lora-hub (Huang et al., 2023) which need to fine-tune about 3-5% neurons, and W-NCFT which need to fine-tune about 30% neurons, our method largely reduces the total number of fine-tuning parameters, proving that it is a parameter-efficient fine-tuning method.

### A.7 Explanation of the definition of "task"

In this work, the five datasets represent the model's knowledge question answering, logical reasoning, harmful replies, and the model's ability to judge paraphrases and sentiment classification. They are quite different from each other, so we can define different datasets to represent one task. In this way, the five datasets represent five tasks respectively. To further verify that the five datasets are quite different, we draw inspiration from Leng & Xiong (2024) which defines task similarity to measure the relationship among tasks. In detail, We transform each task into a feature $\boldsymbol{f}$ that can represent the task and measure the similarity distance of the features. We use the LLAMA2-7B model to get the embedding of each task's corresponding dataset. We perform padding operations for different task embedding dimensions, and average the embeddings of multiple pieces to get the feature represented by the task. The similarity between any two tasks $a$ and $b$ as follows:

$$\text{sim}(a, b) = \frac{\boldsymbol{f}_a \cdot \boldsymbol{f}_b}{||\boldsymbol{f}_a|| \times ||\boldsymbol{f}_b||}$$

By defining features in this way, we can obtain the similarities between the five datasets, and the results are shown in the Fig 5. From this figure, we can see that the similarity values between different tasks do not exceed 0.5. In particular, only SST-2 and QQP have a similarity of 0.42 (because they both belong to classification and have slightly higher similarity), while the similarity values between other datasets are all lower than 0.3. This proves that the similarity between tasks is very small, and the five datasets can be defined as five different tasks. This also verifies the rationality of our task definition.

### A.8 Decision of fine-tuning order and sensitivity analysis

Draw inspiration from (Bell & Lawrence, 2022; Wang & Li, 2024)'s work which design method to analyze the sensitivity of model performance to fine-tuning order by defining different fine-tuning orders, we use the aforementioned similarity matrix to define the distance between tasks and the longest and shortest fine-tuning paths (the distance between tasks is defined here as 1-similarity,

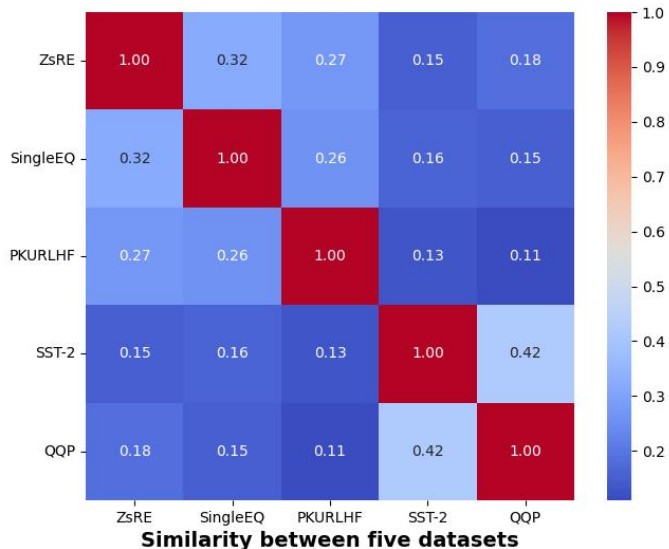

Figure 5: Visualization of similarity of five datasets.

Table 8: Comparison results of different fine-tuning order in multi-task generalization setting.

|  | Average Performance |  | Average Performance |  |
|---|---|---|---|---|
| Maximum Path | 0.6288 |  | 0.6695 |  |
| Minimum Path | 0.6276 | OPT-1.3B | 0.668 | LLAMA2-7B |
| Random Path | 0.6285 |  | 0.6689 |  |
| Main method | 0.6283 |  | 0.6691 |  |

and the longest path is the maximum total distance between tasks passed when all fine-tuning is completed). The specific maximum path and minimum path are shown in the Fig 6. We compare the results of fine-tuning along the maximum path, fine-tuning along the minimum path, fine-tuning along the random path, and the fine-tuning order of our main method, and show them in the Tab 8. We compare the average performance of the models under these four settings (the average performance is measured by the average indicators on the five tasks) and find that the maximum path fine-tuning method has slightly better results than the minimum path, but the significance is weaker ($p < 0.05$, two-sided t-test), while the maximum and minimum path results are not significantly different from the results of our main experiment and random path, which proves that our method has high stability in the choice of fine-tuning order. We conclude the main reason is that there is almost no overlap between task-specific neurons, which makes the fine-tuning parameters of different tasks not interfere with each other and improves the stability of the model to the fine-tuning order.

918
919
920
921
922
923
924
925
926
927
928
929
930
931
932
933
934
935
936
937
938
939
940
941
942
943
944
945
946
947
948
949
950
951
952
953
954
955
956

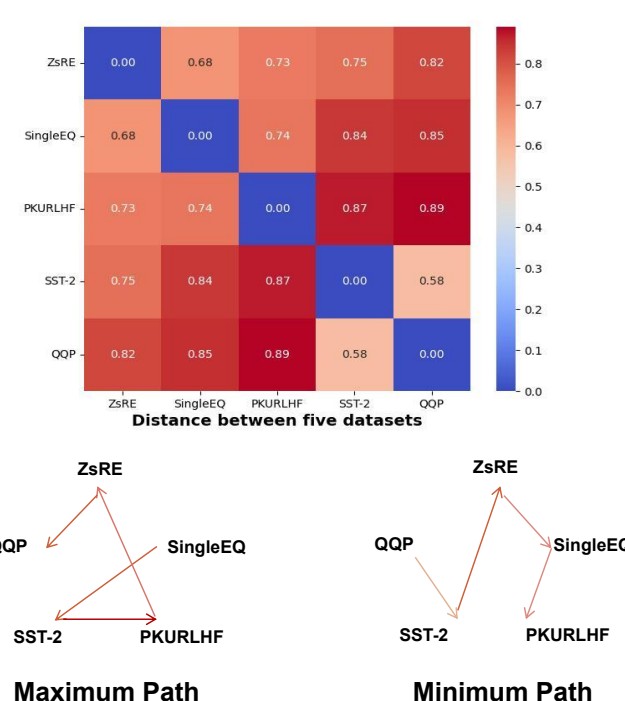

Figure 6: Visualization of distance of five datasets and maximum and minimum path to fine-tune all tasks.
