# OpenReview forum: "Locate-then-Unlearn: An Effective Method of Multi-Task Continuous Learning for Large Language Models"
_ICLR.cc/2025/Conference — Submitted to ICLR 2025_

### Official Review · Reviewer_WAbD · 2024-11-02

**Soundness:** 2
**Presentation:** 3
**Contribution:** 3
**Rating:** 5
**Confidence:** 4

**Summary:**

This paper proposes Locate-then-Unlearn framework to improve LLM continual learning by identifying and selectively unlearning task-specific neurons. By localizing neurons specific to individual tasks and unlearning them, the approach reduces interference between tasks and enhances model efficiency. Experiment results and analyses across multiple datasets demonstrate good performance in both single-task and multi-task settings.

**Strengths:**

1. The proposed approach is novel.
2. The proposed approach is well-motivated.
3. The results show good performance.

**Weaknesses:**

1. One problem of LLM continual learning is catastrophic forgetting, it would be better if the authors can further discuss how the proposed approach handle the issue.
2. For LLM continual learning, it would be better to compare model performance on the benchmark datasets before and after training, so the readers will learn the effect of the learning process on previously learnt task. Currently, the authors just report the performance after training. It would be a plus if the authors can include the model performance before training.
3. The paper claims many times that the proposed approach "significantly" outperforms existing approaches, but it seems the significance test is missing. So, it is hard to justify whether the improvement is significant.

**Questions:**

See weaknesses.

---

> ### Author Response · Authors · 2024-11-29
> **Response to Reviewer WAbD**
>
> Thank you sincerely for your thoughtful comments. We have made every effort to address your concerns in detail as follows.
>
> Besides, we would like to apologize for any inconvenience and any extra efforts for you to check an updated PDF file, which is revised with more accurate descriptions and claims to meet the appropriate standards. We are truly grateful if you take all these matters into consideration and evaluation. Thank you!
>
> >**W1** : ...catastrophic forgetting...
>
> On the one hand, our Locate-then-unlearn method first identifies task-specific neurons, and then subsequent unlearning and relearning are only performed on task-specific neurons for each task. This process effectively deduplicate neurons that have a high impact across multiple tasks, preserving neurons that are critical for one task while minimizing their influence on others. Reducing conflicts between neurons could alleviate the problem of catastrophic forgetting.
>
> On the other hand, we have conducted experiments to further validate this claim. We compare the results of our model under two sequential fine-tuning settings:  (i) **Immediate test**, which means we test the model immediately after training on the target dataset. (ii) **Final test**, which means we test the model finally after training on all datasets. The results in Table 7 show that the performance gap across the first four datasets is much smaller with Locate-then-unlearn compared to the baseline and the ablation variant without locating neurons. This confirms that our method effectively mitigates catastrophic forgetting.
>
> >**W2** : ...before and after training...
>
> To address your concern about comparing results before and after training, we have conducted experiments on all five datasets following the learning sequence outlined in the paper. Specifically, the "Before xx" results correspond to the model fine-tuned sequentially on the previous datasets but not yet on the target dataset. In contrast, the "After xx" results reflect the performance of the model fine-tuned on both the previous datasets and the target dataset in sequence. Since only multi-task generalization is relevant to LLM continual learning, we report the following results under this setting below.
>
> |Metric|Locate-then-unlearn (OPT-1.3B)|Locate-then-unlearn (LLAMA2-7B)|
> |:-|:-:|:-:|
> |**Before ZsRE**|||
> |Specificity|0.2633|0.2884|
> |Generality|0.2281|0.2465|
> |Locality|0.1936|0.2207|
> |**After ZsRE**|||
> |Specificity|0.8594|0.9012|
> |Generality|0.8389|0.8751|
> |Locality|0.7847|0.8201|
> |**Before SingleEQ**|||
> |Acc|0.0775|0.1012|
> |**After SingleEQ**|||
> |Acc|0.2848|0.3127|
> |**Before PKURLHF**|||
> |HarmfulRate|0.4970|0.4655|
> |**After PKURLHF**|||
> |HarmfulRate|0.0135|0.0109|
> |**Before SST-2**|||
> |Acc|0.5110|0.5270|
> |**AfterSST-2**|||
> |Acc|0.8932|0.9245|
> |**Before QQP**|||
> |Acc|0.2133|0.2447|
> |**After QQP**|||
> |Acc|0.7668|0.8192|
>
>
> >**W3** : ...singificance test...
>
> We have performed a significance test using a two-tailed t-test to compare the performance of our Locate-then-unlearn method with that of the best baseline method on each metric.
> The main results are derived from four iterations of bootstrapping. The t-test results show that except for the Locality of ZsRE, the p-values are all less than 0.005 (the highest is 0.00128). While ROME performs the best in Locality, it significantly underperforms compared to Locate-then-unlearn in other metrics. This confirms the statistical significance of our model's performance improvements.

---

### Official Review · Reviewer_UprD · 2024-11-02

**Soundness:** 3
**Presentation:** 3
**Contribution:** 3
**Rating:** 6
**Confidence:** 3

**Summary:**

This paper proposes locate-then-unlearn, to identify and unlearn task-specific neurons in transformer models for multi-task learning. Specifically, the authors propose to identify task-related neurons from top-k important scores for each task, and identify task-specific neurons by removing overlapping neurons shared by different tasks. Finally, the authors fine-tune the selected task-specific parameters with unlearning data, and remove the parameter offset (from fine-tuning) from the original parameters. Experiment results demonstrate that the proposed locate-then-unlearn outperform other knowledge editing baselines on five datasets. They also conduct extensive analysis to demonstrate that the setting of unlearning hyperparameter is important, task-specific neurons work the best on corresponding related tasks.

**Strengths:**

- The method is intuitively reasonable. By detecting task-related and task-specific neurons, the methods make multi-task learning more parameter-efficient.
- The experiment results show that the proposed method can outperform knowledge editing methods such as ROME and MEMIT on five datasets.
- The authors conduct extensive analysis. The unlearning hyperparameter has a best performing peak, while the threshold is rather robust. The task-specific neurons are important, and tuning them on unrelated tasks lead to worse results. Visualization shows that task-specific neurons are well-separated.
- The writing of the paper is quite clear.

**Weaknesses:**

- Since the authors claim that their methods contribute to parameter-efficient multi-task learning, it would be better to compare with methods such as prompt-tuning, lora-hub.
- It would be better if the authors could release their code upon acceptance.

**Questions:**

- How does the method compare with other PEFT methods, such as prompt-tuning, lora-hub?
- It would be better if the authors can discuss for each new task, how many parameters need to be stored (the percentage of the whole transformer model)?

**Details Of Ethics Concerns:**

No ethics concerns are found.

---

> ### Author Response · Authors · 2024-11-29
> **Response to Reviewer UprD**
>
> Thank you sincerely for your insightful feedback. We have made every effort to address your concerns in detail as follows.
>
> Besides, we would like to apologize for any inconvenience and any extra efforts for you to check an updated PDF file, which is revised with more accurate descriptions and claims to meet the appropriate standards. We are truly grateful if you take all these matters into consideration and evaluation. Thank you!
>
> >**W1 & Q1** : ...compare with other PEFT methods, such as prompt-tuning, lora-hub.
>
> Thank you for your constructive advice. We have incorporated two suggested PEFT methods: P-tuning v2 [1] and Lora-hub [2], in comparison with our method and existing baselines on five datasets. The results below show that in multi-task generalization, P-tuning v2 can perform better than F-learning but inferior to W-NCFT and our Locate-then-unlearn, while Lora-hub performs worse than all baselines. This may be because both P-tuning v2 and Lora-hub fail to select the most specific parameters for learning to adapt to different tasks.
> |Multi-task generalization||ZsRE||SingleEQ|PKURLHF|SST-2|QQP|
> |:-|:-:|:-:|:-:|:-:|:-:|:-:|:-:|
> |**Method**|**Specificity↑**|**Generality↑**|**Locality↑**|**Acc↑**|**HarmfulRate↓**|**Acc↑**|**Acc↑**|
> | Lora-hub                   | 0.7221 | 0.6869 | 0.6286 | 0.1898   | 0.1012  | 0.7428 | 0.6095 |
> | P-tuning v2                | 0.7508 | 0.7122 | 0.6463 | 0.2029   | 0.0916  | 0.7542 | 0.6311 |
> | F-learning                 | 0.7383 | 0.6985 | 0.6398 | 0.1932   | 0.0885  | 0.7339 | 0.6013 |
> | W-NCFT                     | 0.7824 | 0.7331 | 0.6559 | 0.2203   | 0.0664  | 0.7857 | 0.6881 |
> | Locate-then-unlearn        | 0.8504 | 0.8277 | 0.7636 | 0.2833   | 0.0144  | 0.8916 | 0.7668 |
>
>
> |Single-task specialization||ZsRE||SingleEQ|PKURLHF|SST-2|QQP|
> |:-|:-:|:-:|:-:|:-:|:-:|:-:|:-:|
> |**Method**|**Specificity↑**|**Generality↑**|**Locality↑**|**Acc↑**|**HarmfulRate↓**|**Acc↑**|**Acc↑**|
> | Lora-hub                   | 0.7606 | 0.7122 | 0.6784 | 0.2110    | 0.0597  | 0.8033 | 0.6953 |
> | P-tuning v2                | 0.7881 | 0.7389 | 0.7002 | 0.2268   | 0.0484  | 0.8196 | 0.7189 |
> | F-learning                 | 0.8018 | 0.7683 | 0.7257 | 0.2242   | 0.0367  | 0.8673 | 0.7580  |
> | W-NCFT                     | 0.7893 | 0.7482 | 0.6884 | 0.2497   | 0.0455  | 0.8110  | 0.7263 |
> | Locate-then-unlearn        | 0.8595 | 0.8391 | 0.7842 | 0.3071   | 0.0103  | 0.9027 | 0.8079 |
>
> >**W2**: release their code
>
> Thank you for your kind reminder. We have uploaded the preliminary code in Supplementary Material.
>
> >**Q2** : ...for each new task, how many parameters need to be stored...
>
> We set the threshold $r_k$ as top 2% in the task-related neuron localization process. After further filtering by task-specific identification, the amount of learnable parameters accounts for approximately 1%-2% of the full model. Specifically, the percentage for ZsRE, SingleEQ, PKURLHF, SST-2, and QQP are 1.43%, 1.72%, 1.09%, 1.50%, and 1.48%, respectively. This demonstrates the parameter-efficiency of our method.
>
> [1] Liu et al. P-tuning v2: Prompt tuning can be comparable to fine-tuning universally across scales and tasks.
>
> [2] Huang et al. Lorahub: Efficient cross-task generalization via dynamic lora composition.

---

### Official Review · Reviewer_Vvor · 2024-11-03

**Soundness:** 3
**Presentation:** 3
**Contribution:** 2
**Rating:** 6
**Confidence:** 3

**Summary:**

This work proposes to unlearn task-specific neurons to enable more efficient multi-task learning. The proposed approach, locate-then-unlearn, first locates task-relevant neurons, deduplicates the neurons that are relevant for multiple tasks, subtracts the corresponding parameters to unlearn false answers, and relearns the objective tasks.

The proposed approach shows higher performance than five baseline methods on five datasets in the multi-task learning scenario, and improvements on three out of five datasets in the single-task setting (with the remaining two getting close performance with direct full fun-tuning).

**Strengths:**

* This work presents solid experimental results and ablation studies, showing the effectiveness of each component.

* The main novelty lies in finding task-specific neurons for more efficient re-learning.

**Weaknesses:**

* The causal effect between the neurons and different tasks requires more careful examination. Although the approach shows empirical improvement,  the assumption that neurons are task-specific might require further evidence.

* This approach is dependent on the selected data used to locate neurons (section 3.2.) and unlearn tasks (section 3.3). The authors provide several ablation studies on the method components, while it may be worthwhile to conduct ablations on the choice of data.

**Questions:**

* What is the definition of “tasks” in this work? The distribution of data, even under the same tasks, seems to affect the selection of task-specific neurons. Thus, whether improvement will transfer to different datasets from the same tasks is unexamined.

* How are the toy datasets constructed in section 3.2? Are they a subsample of the underlying task data?

* Regarding the design of false answers in section 4.3., the false knowledge can be presented in multiple forms than what’s already in the dataset, which could thus not capture the potential wrong knowledge and influence the effect of unlearning.

*Typos*

* Title: Continuous -> Continual

* L409-410: implement -> implementing; this sentence is difficult to parse.

---

> ### Author Response · Authors · 2024-11-29
> **Response to Reviewer Vvor (1/2)**
>
> Thank you sincerely for your valuable comments. We have made every effort to address your concerns in detail as follows.
>
> Besides, we would like to apologize for any inconvenience and any extra efforts for you to check an updated PDF file, which is revised with more accurate descriptions and claims to meet the appropriate standards. We are truly grateful if you take all these matters into consideration and evaluation. Thank you!
>
> >**W1** : ...the assumption that neurons are task-specific might require further evidence.
>
> In response to your valuable feedback, we have conducted experiments to verify the assumption of task-specific neurons by exploring the behavior of task-specific neurons under different learning scenarios. Specifically, we implement the following settings for comparison: (i) **Within-task learning**: Fine-tuning task-specific neurons on their respective task datasets; (ii) **Cross-task learning**: Fine-tuning task-specific neurons using datasets from other tasks instead of their corresponding datasets; (iii) **Full parameter fine-tuning**: Fine-tuning all parameters of the model (not just task-specific neurons) on each task dataset.
>
> The results below (also shown in Table 3) highlight that task-specific neurons achieve better performance when fine-tuned on their respective datasets compared to the other two settings, especially in tasks such as ZsRE, SingleEQ, and PKURLHF. For instance, when fine-tuning ZsRE-specific neurons on the ZsRE dataset, the accuracy reached around 89%. In contrast, QQP-specific neurons fine-tuned on ZsRE only resulted in accuracies below 40%. Additionally, fine-tuning task-specific neurons on their respective datasets is comparable to or even better than full parameter fine-tuning, further indicating the effectiveness of task-specific identification.
> ||ZsRE↑|SingleEQ↑|PKURLHF↓|SST-2↑|QQP↑|
> |-|:-:|:-:|:-:|:-:|:-:|
> |Full parameter fine-tuning|81.08%|32.75%|1.18%|**92.41%**|**83.29%**|
> |ZsRE|**88.71%**|20.74%|6.60%|62.16%|51.56%|
> |SingleEQ|64.55%|**32.96%**|10.04%|58.98%|50.28%|
> |PKURLHF|58.12%|15.85%|**1.03%**|60.34%|55.79%|
> |SST-2|37.31%|8.27%|18.76%|_92.30%_|71.47%|
> |QQP|39.17%|7.78%|19.52%|81.65%|_82.88%_|
>
> >**Q1** : the definition of "task"...different datasets from the same tasks is unexamined.
>
> Thank you for your constructive suggestions.
>
> **Regarding the definition of "task"**, in our main experiments, the five datasets are carefully chosen to represent distinct aspects of the model's capabilities, including knowledge-based question answering, logical reasoning, harmful replies, paraphrase judgment, and sentiment classification. These datasets are quite different in their nature, which justifies treating them as representing separate tasks.
>
> To provide further validation, we draw inspiration from [1] and convert the tasks into embeddings to calculate the correlation scores between datasets. The results in Figure 5 show that the maximum correlation score between any two datasets does not exceed 0.5, further supporting our assumption that these datasets capture different tasks.
>
> **Regarding the transferability to different datasets within the same task**, we have conducted additional experiments using two new datasets: (i) **LogiQA** [2]: another logical reasoning dataset, has a similarity score of 0.81 with SingleEQ, suggesting they belong to the same task; and (ii) **MNLI** [3]: another dataset for semantic relationship judgment, has a similarity score of 0.74 with SST-2, indicating they are in the same task category. The following results on LogiQA and MNLI show that our Locate-then-unlearn still achieves better performance than other baselines, demonstrating the robustness and transferability of our method to new datasets.
> ||OPT-1.3B||LLAMA2-7B||
> |:-|:-:|:-:|:-:|:-:|
> |**Method**|**LogiQA**|**MNLI**|**LogiQA**|**MNLI**|
> |Directly fine-tuning|0.4469|0.7817|0.4754|0.8263|
> |ROME|0.3596|0.7447|0.3868|0.7835|
> |MEMIT|0.4659|0.8012|0.4948|0.8424|
> |F-learning|0.5062|0.8341|0.5334|0.8630|
> |W-NCFT|0.5326|0.8738|0.5658|0.8972|
> |Locate-then-unlearn|0.5664|0.9065|0.6019|0.9283|

---

> > ### Author Response · Authors · 2024-11-29
> > **Response to Reviewer Vvor (2/2)**
> >
> > >**W2 & Q2** : toy dataset construction and ablation study.
> >
> > Thank you for the insightful question. Each toy dataset we used is actually a random subsample of training datasets of each task. To assess whether different constructions of the toy dataset significantly impact the model performance, we further compare several settings to select the toy dataset: (i) **Fixed number of samples**, experimented with 500 and 1000 random samples for each task; (ii) **Proportional sampling**, a proportionate number of random samples based on the scale of the original data, experimented with proportions of 5% and 10% for each task. We perform experiments on multi-task generalization and single-task specialization under each same setting. The following results (also shown in Table 4) indicate that our method is relatively robust and not heavily dependent on the specific construction of the toy dataset.
> > |||ZsRE||SingleEQ|PKURLHF|SST-2|QQP|
> > |-|:-:|:-:|:-:|:-:|:-:|:-:|:-:|
> > |**Setting**|**Specificity↑**|**Generality↑**|**Locality↑**|**Acc↑**|**HarmfulRate↓**|**Acc↑**|**Acc↑**|
> > |500 cases|0.8501|0.8278|0.7633|0.2833|0.0145|0.8914|0.7668|
> > |1000 cases|0.8504|0.8277|0.7636|0.2833|0.0144|0.8916|0.7668|
> > |5%|0.8503|0.8276|0.7634|0.2831|0.0145|0.8917|0.7666|
> > |10%|0.8505|0.8277|0.7638|0.2833|0.0143|0.8917|0.7668|
> >
> > >**Q3** : ...false knowledge can be presented in multiple forms...
> >
> > Thank you for your thoughtful comments. We have conducted the analysis to assess how presenting multiple false answers influences the performance. For the SingleEQ dataset, which originally contained no false answers, we use GPT-4 to generate four plausible incorrect answers along with their intermediate reasoning processes that differed from the correct answer. For the ZsRE and PKURLHF datasets, which originally had one false answer per question, we added three additional incorrect answers based on the original one. While for SST-2 and QQP with fixed categories, it is not suitable to augment irrelevant false answers.
> >
> > The following results (also shown in Table 6) suggest that using multiple false answers does not significantly improve the performance, and in some cases, may even degrade it. Given the additional time required for generating multiple answers using LLMs, we determine to maintain a simpler setup, where each question is associated with only a single false answer.
> > |Multi-task generalization||ZsRE||SingleEQ|PKURLHF|
> > |:-|:-:|:-:|:-:|:-:|:-:|
> > |**Method**|**Specificity↑**|**Generality↑**|**Locality↑**|**Acc↑**|**HarmfulRate↓**|
> > |F-learning|0.7383|0.6985|0.6398|0.1932|0.0885|
> > |W-NCFT|0.7824|0.7331|0.6559|0.2203|0.0664|
> > |Locate-then-unlearn (one false answer)|0.8504|0.8277|0.7636|0.2833|0.0144|
> > |Locate-then-unlearn (four false answers)|0.8443|0.8215|0.7578|0.2875|0.0167|
> >
> > |Single-task specialization||ZsRE||SingleEQ|PKURLHF|
> > |:-|:-:|:-:|:-:|:-:|:-:|
> > |**Method**|**Specificity↑**|**Generality↑**|**Locality↑**|**Acc↑**|**HarmfulRate↓**|
> > |F-learning|0.8018|0.7683|0.7257|0.2242|0.0367|
> > |W-NCFT|0.7893|0.7482|0.6884|0.2497|0.0455|
> > |Locate-then-unlearn (one false answer)|0.8595|0.8391|0.7842|0.3071|0.0103|
> > |Locate-then-unlearn (four false answers)|0.8537|0.8371|0.7819|0.3103|0.0135|
> >
> > > _Typos_
> >
> > Thank you for your careful proofreading. We have fixed the typos in the revised PDF.
> >
> > [1] Leng & Xiong. Towards understanding multi-task learning (generalization) of llms via detecting and exploring task-specific neurons.
> >
> > [2] Liu et al. Logiqa: A challenge dataset for machine reading comprehension with logical reasoning.
> >
> > [3] Wang et al. Glue: A multi-task benchmark and analysis platform for natural language understanding.

---

### Public Comment · ~Nasim_Nour_Mohammed1 · 2024-11-15
**This paper violates the double-blind review process and is questionable**

**Dear reviewers and chairs,**

There are some serious concerns about the academic integrity of this paper under review.

Some people on Twitter have raised issues about the authors violating the double-blind review process by revealing their submission ID and author list on a Chinese social media: [removed]

Upon reading the paper, I believe it exhibits more substantial academic issues. Below, I outline my arguments:

> 1. Lack of Originality in Neuron Locating Method

The paper claims to propose a "two-stage locating method" for identifying task-specific neurons (Line 90). However, this approach is very similar to the method presented by Chen et al. [1], which also involves a two-stage process—first identifying **knowledge-related** neurons, and then filtering those that store factual knowledge **transcending** language. The authors of this paper first locate “**task-related**” neurons, then filter task-specific neurons, because “some task-related neurons may exhibit high logit scores **across multiple tasks**. (Line 70)” The authors replace the concept of "language" with "task" without citing Chen et al.

Furthermore, the neuron identification method (Eq.2) in this paper is **identical** to Eq.4 of Tang et al. [2]. The authors replace "language" in the original paper with "task" once again. Here is Equation 4 from Tang et al.:

$ p_{i,j}^k = \mathbb{E} \left( \mathbb{I}(	ext{act fn}(	ilde{h}^i W_1^i)_j > 0) \mid 	ext{language } k ight), $

The authors state, "the definition of a neuron follows Tang et al. (Line 143)," but they do not attribute Eq.2 to Tang et al.'s contribution. Such wording seems intended to mislead reviewers unfamiliar with Tang et al.

> 2. Continual Learning vs Lifelong Editing

At first, I did not understand why they called this model editing method continual learning. However, upon closer inspection, particularly of Eq.5, it becomes clear that the subtraction between parameters of two models is a common practice in lifelong model editing (see Eq.3 in Wang et al. [4]). Nevertheless, the authors do not cite any relevant lifelong editing literature, raising concerns about proper attribution.

> 3. The Style of Figures Demonstrates That the Authors Have Read the Papers I Listed

This paper does not relate to attention modules at all, yet the authors still include attention modules in Figure 1. Why? This figure closely resembles Figure 1 from Meng et al. [5], with the authors largely following the figure style from this classic paper on model editing. Additionally, the top-right part of Figure 1 in this paper uses elements similar to Figure 1 of Tang et al., and the structure of Figure 1(a) and (b) in this paper is similar to that of Figure 2 in Chen et al. (Figure 1(a) and (b) illustrate the two-stage neuron identification and recap that this two-stage process was proposed by Chen et al.)

While reusing classic figures is not inherently problematic, the resemblance suggests that the authors have read these papers but still underplayed their contributions in this paper.

> Confusing Equations

Equation 3 in the paper uses a superscript in the summation notation that is unclear. I wonder whether the reviewers understand this superscript—is it n or k?

If it is k, as k represents a cardinal task index, the meaning of "i < k" is ambiguous.

If it is n, the summation term is a constant! It is strange to subtract a constant in this metric. However, based on the related works cited, I believe this formulation may have been borrowed from work in mechanistic interpretability, such as the circuit discovery methods in Lv et al. cited in this paper, which calculate the logit difference and normalize it with the average logits over the vocabulary to measure activation differences between two inputs. It is an odd mistake to reuse such a measure for probability differences in this paper because averaging the probability difference over the distribution results in a constant.

In summary, I believe that the paper demonstrates substantial overlap with prior works in both methodology and presentation, with some clear mistakes and without proper citations or acknowledgment.

I appreciate the reviewers' efforts, but for the integrity of the ICLR, I hope the reviewers and chairs will consider the issues I have raised and re-evaluate the paper.

**Given the violation of the double-blind process, this paper should perhaps be desk-rejected directly.**

[1] Chen et al. Journey to the Center of the Knowledge Neurons: Discoveries of Language-Independent Knowledge Neurons and Degenerate Knowledge Neurons

[2] Tang et al. Language-Specific Neurons: The Key to Multilingual Capabilities in Large Language Models

[3] Geva et al. Transformer Feed-Forward Layers Are Key-Value Memories

[4] Wang et al. LEMoE: Advanced Mixture of Experts Adaptor for Lifelong Model Editing of Large Language Models

[5] Meng et al. Locating and Editing Factual Associations in GPT

---

> ### Author Response · Authors · 2024-11-26
> **First Author Response**
>
> Thank you for your attention to our paper. We have revised our manuscript to clarify the raised concerns and elaborate with more accurate claims. Please kindly check the updates.
>
> As to the concern regarding to possible double-blind violation, it was my first submission as the first author, I was so inexperienced and I did not ask for consent from any of my co-authors. We have consulted the PC chairs about this matter, and are waiting for their decision.
>
> Sincerely,
>
> First Author of Submission 3322

---

> > ### Public Comment · ~Nasim_Nour_Mohammed1 · 2024-11-27
> > **I can not believe that the authors have rewritten the paper without any apology**
> >
> > To Songshi Liang
> >
> > It is not justifiable to violate the reviewing guidelines, even for a first-time submission. Once your identity and submission ID are revealed, the review process is no longer blind, compromising its fairness. It is important to acknowledge that you undermined the integrity of the review.
> >
> > Additionally, your academic dishonesty cannot be overlooked, even if you update the PDF. Considering that this is your first submission, I tried not to use the word 'plagiarism,' even though my previous comment was not taken seriously, as you posted rebuttals WITHOUT ANY APOLOGY. It is evident that you have revised all the points from my comment, removed almost all of your claimed contributions and novelty, and corrected the credit to the original papers. Your act is a serious matter, regardless of any revisions you make.
> >
> > In conclusion, I trust that the conference chairs will make a fair and thorough decision regarding this submission. I also hope the reviewers will re-evaluate the current version of this paper, as the authors have essentially rewritten the entire paper according to my comments. The latest version offers very limited original contributions.
> >
> > By the way, the link in my previous comment disclosing your violation of the double-blind review process appears to have been deleted. I'm not sure if OpenReview doesn't allow posting links, so I'm sharing the link here again: https://x.com/saulgoodma1995/status/1856616262039957581

---

> > > ### Author Response · Authors · 2024-11-28
> > > **Thank you again for your comments**
> > >
> > > Thank you for sharing your perspective. I'd like to respond to your concerns one by one.
> > >
> > > 1. Regarding the possible double-blind violation, as I said, I have brought this matter to the attention of the PC Chairs, and am waiting for their decision. Since we both trust the chairs will have a fair and thorough decision, we shall just wait for their final call.
> > >
> > > 2. As to the contribution, since we have discussed the comments in the public thread, the reviewers will receive the discussion notification as well. I am sure that your comments have reached them already. Perhaps it would be best to leave the reviewers to make their own evaluations and form independent judgments.
> > >
> > > 3. In the earlier version, I made efforts to credit the authors of my inspirations by citing the relevant reference papers consistently. While I do not believe this constitutes "academic dishonesty", l agree that striving for higher standards is always beneficial. That's why I reorganized the paper, added a preliminary section, and refined the contribution statement to avoid any over-claims. However, I must clarify that the paper was not "completely rewritten": the revisions were made to ensure alignment with appropriate standards. You can check the contribution statements in detail by comparing the two versions.
> > >
> > >
> > > | **Before Revision** | **After Revision** |
> > > |-|-|
> > > | - We propose locate-then-unlearn, a novel framework that facilitates efficient and continual learning for LLMs across multiple tasks.  | - We propose Locate-then-unlearn, a new framework that facilitates efficient and continual learning for LLMs across multiple tasks.  |
> > > | - We develop a two-stage locating method to accurately identify task-specific neurons by assessing their importance and isolating those critical to each task.  | - We develop a new locating method to accurately identify task-specific neurons by assessing their importance and isolating those critical to each task.  |
> > > | - We design two experimental settings for comprehensive evaluation: single-task specialization and multi-task generalization. Experimental results show that our method achieves significant improvements in both settings, demonstrating an effective balance between performance and efficiency.  | - We design two experimental settings for comprehensive evaluation: single-task specialization and multi-task generalization. Experimental results show that our method achieves significant improvements in both settings, demonstrating an effective balance between performance and efficiency.  |
> > >
> > > Thank you again for your input. This work is my first submission and takes me almost a year to complete. It means a lot to me, and I have taken your advice seriously by revising the paper accordingly to ensure it meets the appropriate standards. I trust that the PC Chairs and reviewers will arrive at a fair and sufficiently informed decision.
> > >
> > > First Author of Submission 3322

---

### Meta-Review · Area_Chair_vmRV · 2024-12-18

**Metareview:**

This paper proposes a locate-then-unlearn framework to first identify task-specific neurons for each task and then unlearn the false information embedded in these neurons, followed by a relearning process. The proposed framework effectively handles multi-task learning where task-agnostic neurons are kept but only those that are most functional towards a task are being updated. Comprehensive experiments on both single-task and multi-task settings are conducted, demonstrating the advantage of the proposed method over existing baselines.

Strengths:
- The method specifically targets efficient multi-task learning by identifying a small number of neurons responsible for a specific task and only updating these neurons. This motivation is reasonable and the solution is effective.
- Extensive experiments are performed, including ablation studies and in-depth analysis, showcasing the advantage of the proposed framework in both single-task and multi-task settings, compared with direct fine-tuning and baseline editing models. Complexity analysis is also provided to demonstrate its efficiency.

Weaknesses:
- The paper has gone through a non-trivial structural and content update, compared to its first submission. This update has led to a change of scope, contribution and methodology. Due to such changes, it is advised that the paper goes through a new round of review before making an informed decision.
- The novelty is a bit limited, after comparing with related work. While the authors have adjusted the contribution statements and restructured the methodology by adding a preliminary section, it weakens the contribution in terms of methodology, as both the localization of neurons and unlearning strategies have been proposed. The adaptation of these strategies into a multitask setting is interesting, but not sufficiently novel.
- The identification of task-specific neurons relies on intervention and observing the difference using a toy dataset. Yet, the sensitivity towards the selected data is not fully examined.

**Additional Comments On Reviewer Discussion:**

- The reviewers have raised questions regarding the causal effect between the identified neurons and different tasks, as well as the effect of different data sampling. In response, the authors conducted additional experiments verifying the task-specific neurons bring most performance gain when finetuned on the respective task dataset. To examine the robustness towards data sampling, additional experiments with different sampling strategies are performed. These additional experiments are helpful in strengthening the paper's claim.
- Reviewers also raised questions regarding the efficiency, such as number of trainable parameters and comparisons with PEFT. The authors added experiments comparing with PEFT baselines. In addition to the experiments, more discussions are needed to explain the tradeoff between number of trainable parameters and performances, as the comparison in terms of parameter size is not given.
- Another interesting question relates to continual learning and catastrophic forgetting. To answer this question, the authors conducted experiments varying the time (checkpoints) for evaluation. Interestingly, the proposed method is beneficial in avoiding forgetting.

Overall, the authors have made efforts addressing the reviewers' questions which could further strengthen this paper.

---

### Decision · Program_Chairs · 2025-01-22

Reject